# Kinome inhibition states and multiomics data enable prediction of cell viability in diverse cancer types

**Matthew E. Berginski** [1], **Chinmaya U. Joisa** [2], **Brian T. Golitz** [3], **Shawn M. Gomez** [1,2]*

**1** Department of Pharmacology, University of North Carolina at Chapel Hill, Chapel Hill, North Carolina, United States of America, **2** Joint Department of Biomedical Engineering, University of North Carolina at Chapel Hill and North Carolina State University, Chapel Hill, North Carolina, United States of America, **3** Eshelman Institute for Innovation, University of North Carolina at Chapel Hill, Chapel Hill, North Carolina, United States of America

☯ These authors contributed equally to this work.
* smgomez@unc.edu

**Data Availability Statement:** All of the code written to support this paper is available through github (https://github.com/gomezlab/kinotype_viability) along with a walkthrough explaining where to find

## Abstract

Protein kinases play a vital role in a wide range of cellular processes, and compounds that inhibit kinase activity emerging as a primary focus for targeted therapy development, especially in cancer. Consequently, efforts to characterize the behavior of kinases in response to inhibitor treatment, as well as downstream cellular responses, have been performed at increasingly large scales. Previous work with smaller datasets have used baseline profiling of cell lines and limited kinome profiling data to attempt to predict small molecule effects on cell viability, but these efforts did not use multi-dose kinase profiles and achieved low accuracy with very limited external validation. This work focuses on two large-scale primary data types, kinase inhibitor profiles and gene expression, to predict the results of cell viability screening. We describe the process by which we combined these data sets, examined their properties in relation to cell viability and finally developed a set of computational models that achieve a reasonably high prediction accuracy ($R^2$ of 0.78 and RMSE of 0.154). Using these models, we identified a set of kinases, several of which are understudied, that are strongly influential in the cell viability prediction models. In addition, we also tested to see if a wider range of multiomics data sets could improve the model results and found that proteomic kinase inhibitor profiles were the single most informative data type. Finally, we validated a small subset of the model predictions in several triple-negative and HER2 positive breast cancer cell lines demonstrating that the model performs well with compounds and cell lines that were not included in the training data set. Overall, this result demonstrates that generic knowledge of the kinome is predictive of very specific cell phenotypes, and has the potential to be integrated into targeted therapy development pipelines.

the code relevant to each part of the paper. We have also made all of the model validation results available through zenodo (https://doi.org/10.5281/zenodo.6323686).

**Funding:** This work was supported by the following grants to SMG from the National Institutes of Health - U24DK116204 (NIDDK), U01CA238475 (NCI), R01CA233811 (NCI). (https://www.nih.gov/). The funders had no role in study design, data collection and analysis, decision to publish, or preparation of the manuscript.

**Competing interests:** The authors have declared that no competing interests exist.

## Author summary

Being able to predict how a patient's tumor will respond to a specific drug treatment is a core goal in the field of precision oncology. An emerging trend in targeted therapies is a focus on protein kinases, a family of over 500 proteins that form an integrated communication network that plays a central role in the development and progression of nearly all cancers. Despite the growing importance of these drugs in the oncologist's therapeutic toolbox, our ability to predict the response of a tumor to a given treatment is poor. To see if we could improve our ability to predict a cancer's response to kinase inhibitor treatment, we leveraged a large experimental dataset that quantifies the effect of these drugs on the kinases. Using these kinase inhibition state data within machine learning models, we found that we could predict the response of cancer cell lines representing over 27 cancer types with high accuracy. Including cell line-specific gene expression data that could be gathered in a clinical setting further improved the accuracy of predictions. Together, these results suggest that knowledge of the inhibition state of the kinome has significant potential to improve our ability to design and deliver more effective targeted cancer treatments.

## Introduction

While chemotherapy remains a mainstay in cancer treatment, the use of targeted therapies clearly holds significant promise, with their use leading to improved outcomes in a variety of cancers [1,2]. Examples include the use of imatinib (Gleevec) for chronic myelogenous leukemia, crizotinib and other anaplastic lymphoma kinase (ALK) inhibitors for non-small-cell lung cancers, and trastuzumab and lapatinib for ERBB2/HER2 amplified breast cancers [3–8]. Together with the potential to reduce toxicity and associated side effects, the development of targeted therapies has gained increasing momentum over the last two decades [9,10].

Since the development of imatinib, protein kinases have emerged as a primary focus for targeted therapy development [11–14]. Kinases are a ~500-member enzyme family that catalyzes the transfer of phosphate groups from ATP to specific substrates. Integrated into a complex network of interactions defined as the kinome, kinases regulate information transfer across a myriad of cellular processes including growth, proliferation, differentiation, motility, and apoptosis [15]. Linked to its role in this wide array of functions, dysregulation of one or more members of the kinome is directly implicated in numerous pathologies, especially cancer [16]. Modulation of kinase activity through targeted inhibition has been the primary therapeutic approach to date and as of 2021, over 85 kinase inhibitors have been clinically approved worldwide, though only targeting 42 kinases from the 21 kinase families [17], highlighting the opportunity for further advancement of this large family of druggable targets.

Recent work characterizing kinome behavior in response to targeted kinase inhibitor therapies has established that the kinome is a highly dynamic system, with significant ramifications in our understanding of drug resistance, adaptive reprogramming and the broader design of effective therapies [18–23]. Underlying these investigations of kinome dynamics are the advancement of proteomic approaches that enable the characterization of protein kinome behavior in response to perturbation en masse, allowing characterization of changes not just to the kinase to which the inhibitor was designed, but also across the entire kinome [24,25]. However, while providing transformative insight into how these targeted therapies interact with and modify cellular systems, our understanding of kinome changes and the resulting downstream cellular changes is still lacking.

Given the potential of targeted therapies and the potential to quantitatively assess their effect on the protein kinome, in this work we sought to establish a predictive framework that links the behavior of the kinome as defined by "kinase inhibition states" with a downstream phenotype—in this instance, cell viability. Enabling this effort is recent work by Klaeger et al., who conducted a comprehensive investigation using a proteomic kinobead approach, establishing a target landscape for 229 kinase inhibitors across a wide range of compound concentrations [26]. This work was conducted using a lysate mixture derived from four cell lines which provided a broad representation of the kinome. The results from Klaeger et al. show that many kinase inhibitors have broad target promiscuity and that the kinases targeted by each inhibitor also varies on the basis of the specific compound concentration. Throughout the rest of this paper, we will use the phrase kinase inhibition state to indicate the specific set of kinases targeted by a given compound and to what degree each kinase is inhibited at each concentration. In addition, we utilized the extensive data available via the Broad Institute's Cancer Cell Line Encyclopedia (CCLE) [27], including the PRISM (Profiling Relative Inhibition Simultaneously in Mixtures) highly multiplexed cell viability assay, along with accompanying multi-omics data (gene expression, copy number variation, proteomics and gene essentiality) from the Cancer Dependency Map. These data consist of cell viability measurements for 499 cell lines across 1448 drugs, transcriptomic profiles for 1389 cell lines, whole proteomic profiles for 375 cell lines, whole genome copy number variation for 1750 cell lines and CRISPR-KO genetic dependency scores for 1054 cell lines. While predictive models for drug-induced cell viability have been built using various strategies [28–31], most have focused on using baseline and drug-perturbed transcriptomic data to make predictions on the sensitivity of cancer cell lines to drugs. Drug-target interaction data like kinome profiles are relatively underutilized in these approaches, but have been shown to have predictive power in smaller datasets [32].

Here, we describe a framework that integrates kinome profiling data with general multi-omics, and build tree-based regression models to predict cell viability for 480 cancer cell lines across 230 kinase inhibitors with high accuracy ($R^2 = 0.79$). Integrating nearly half a million data points, we find that kinome inhibition profiles have by far the greatest predictive power of any single data set. While not highly predictive on its own, baseline transcriptomic data does significantly enhance prediction accuracy, "tuning" the model to individual cell lines. Remarkably, adding in other multi-omics data does not significantly increase the quality of predictions. As the model enables prediction of complete dose-response curves, we experimentally validate predictions for over two dozen compounds on two breast cancer cell lines and find strong agreement for most compounds tested. These results suggest that the link between kinotype and phenotype is significant, with sufficient power to enable the prediction of cell viability and likely other cellular phenotypes as well. Along with integration of transcriptional data, these predictive models can greatly enhance our understanding of adaptive kinome reprogramming and drug resistance while facilitating the development of future targeted therapy regimes.

## Results

This work is divided into three parts. We start by describing how we processed and organized the data sets used to build predictive models of cell viability related to a set of kinase inhibitors. Next, we describe the methods we used to select which features and data sets to include in these models and apply a set of modeling methods to the organized data. Finally, we make a set of cell viability predictions and then experimentally test these predictions in a panel of breast cancer cell lines.

## Linking kinome inhibitor states with cancer cell viability

There are two primary data sources that we needed to process and combine in order to link kinotype with phenotype and build a model to predict the cell viability effects of kinase inhibitors. The first of these data sources is the large-scale PRISM cell viability screening effort. The PRISM data collection consists of a set of cell line viability measurements following exposure to a wide range of compounds [33] (Fig 1A). These compounds span multiple different target classes, but in this work we have focused on a specific subset of kinase inhibitors that have been independently assayed using the kinobead/MS-based method. This approach determines

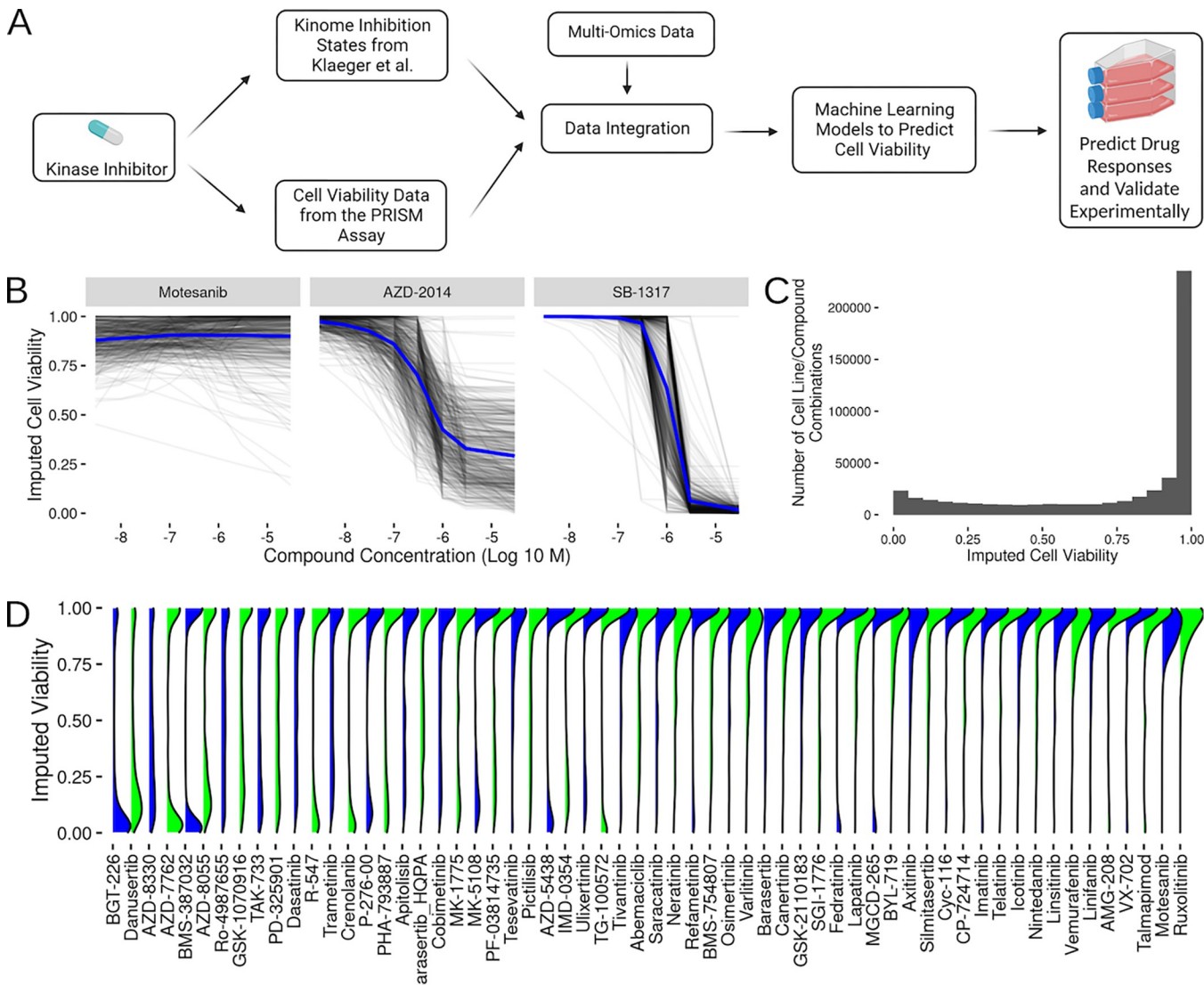

**Fig 1. Study Design Overview and Imputation of Cell Viability from PRISM.** (A) Flow chart showing data source collection, integration and modeling strategy. (B) Sample imputed cell viability curves for all assayed cell lines (gray underlying lines) and corresponding average imputed cell viability response (blue line) for three compounds showing low changes (Motesanib), medium level changes (AZD-2014) and high changes (SB-1317) in cell viability. (C) Overall distribution of cell viability values imputed at Klaeger et al compound concentrations. (D) Distribution of imputed cell viability across all concentrations for a selection (60 out of 168) sampled evenly across the average imputed cell viability effect of the compounds present in both PRISM and the Klaeger et al set. The blue and green color scheme does not indicate anything about the underlying data and is meant to act as a visual aid for differentiating between adjacent curves. (Panel A was created with BioRender.com).

the precise kinase targets as well as the magnitude of their inhibition in response to different concentrations of the inhibitors [26]. Given that the compounds used in Klaeger et al. are all well known kinase inhibitors, most of the proteins that appear in the assay results are either known kinases or closely associated proteins. As such, we'll refer to the data originating from the Klaeger et al. result as "kinase inhibition states."

The primary challenge with combining these data sets is a lack of overlap between some of the concentrations used in the PRISM assay and those used by Klaeger et al. To overcome this problem, we used the viability curve fits provided by the PRISM database and imputed cell viability values for all of the concentrations used by Klaeger et al (Fig 1B). These cell viability results are represented as a value from 0–1, with 0 indicating complete cell death and 1 indicating no effect on cell viability. As expected, a majority of the treatments yielded little change in cell viability (Fig 1C). The distribution of cell viability values within each individual compound shows that while many of the compounds have minimal effects on cell viability, some compounds show a much wider range of viability effects (Fig 1D).

After combining the PRISM and Klaeger et al. data sets, we have 168 compounds which have been assayed across 480 cell lines. We imputed the cell viabilities at each of the 8 concentrations used in the Klaeger et al. work, yielding about half a million treatment combinations across combinations of cell line, compound and concentration. With this data set, we also integrated the gene expression data available through the Cancer Cell Line Encyclopedia [34]. These gene expression values (log2 TPM values with a pseudocount of 1) were available in all but four of the 480 lines used in the PRISM compound screens. Following the integration of gene expression, we next examined how well single kinase inhibition and gene expression values were correlated with cell viability.

## Cell viability after treatment with kinase inhibitors shows mild correlation with kinase inhibition state

We investigated the relationship between kinase inhibition states (~520 proteins) and gene expression values with inhibitor-induced cell viability. To do this, we took each individual kinase inhibition state and gene expression value (~21,000 TPM values) and calculated the Pearson's correlation coefficient with the imputed cell viabilities (Fig 2A and 2B). The kinase inhibition states from Klaeger et al. are represented as a value lying mostly between zero and one, where zero indicates a fully inhibited kinase and values of one or above indicate that a kinase isn't inhibited. These correlations were in general significantly lower for the gene expression values, while the kinase inhibition state values showed both a higher average correlation and greater variance (Fig 2C). This was not unexpected as the gene expression values are all characterized in unperturbed cell lines. Thus, as cell viability changes the gene expression values remain fixed, and any variation across gene expression must be correlated with broad changes in drug response between the cell lines. The examination of single correlation values gives a picture of how well single expression or inhibition states are related to the cell viability phenotype.

While single features with correlation coefficient values in the ~0.3 range (the highest value observed in the kinase inhibition state data) will not produce sufficiently predictive models, the integration of multiple features may provide greater power. As such, we next sought to use the correlation values for feature selection. The most obvious way to use the correlation values is to put all the potential features (in this case, kinase inhibition state and gene expression) in correlation rank order and then select the top-X number of features for model inclusion. This produces differing sets of feature class counts and ratios depending on the number of features selected (Fig 2D right). Interestingly, the top ~350 features all come from the kinase inhibition

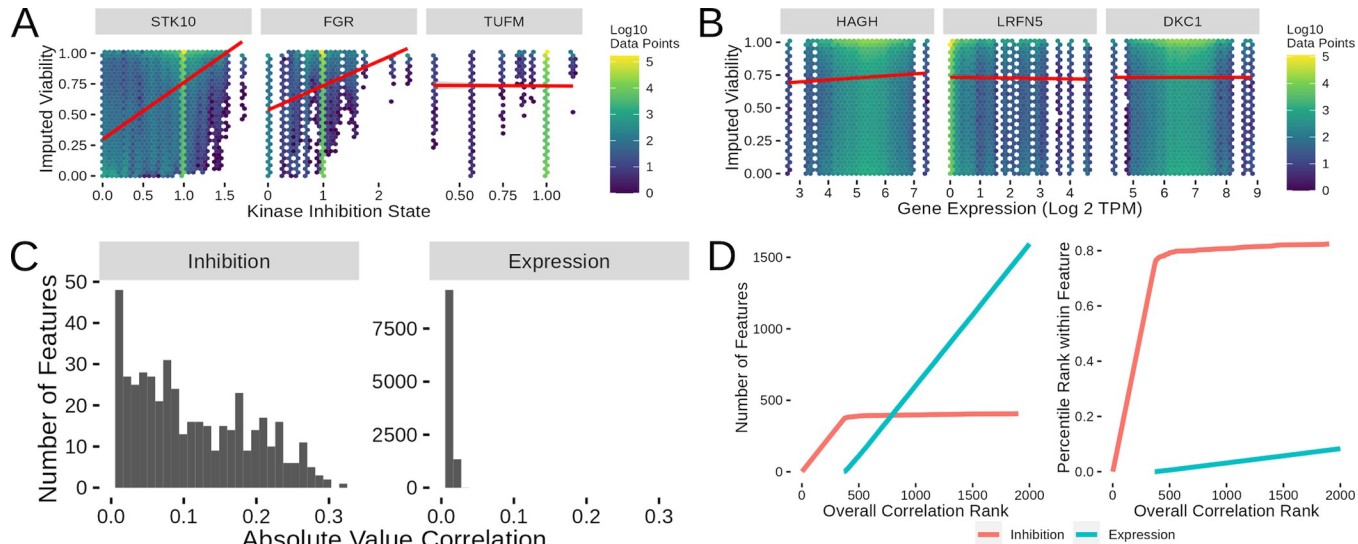

**Fig 2. Single Feature Correlations Across Kinase inhibition and Gene Expression.** (A) Sample kinase inhibition state versus imputed viability heatmap plots showing inhibition states with high (STK10), medium (FGR) and low (TUFM) correlation values. (B) Sample gene expression versus imputed viability heatmap plots showing genes with high (HAGH), medium (LRFN5) and low (DKC1) correlation values. (C) Overall distribution of correlations between kinase inhibition states and gene expression levels. (D) Plots showing what order classes of features are selected from the inhibition and expression correlations. The number of features from each class (left) selected at a given rank value and the percentage of the possible features (right) selected at a given feature selection rank cutoff.

states, with gene expression then starting to be included into the list after the first 350 features. As an alternative method to visualize the same selection process, we plotted what percent of a given feature class is included in the top list for the top 2000 features (Fig 2D left). This alternative view of the feature selection process shows that ~80% of the inhibition states are included in the model before gene expression starts to be included. This indicates that nearly all of the inhibition states are more highly correlated than the gene inhibition states and will thus be the sole factor utilized in lower feature count models. Extending the feature list visualization to include lists greater than 2000 show that remaining inhibition states are slowly included as the top feature list expands (S1A Fig). This analysis of the structure of the single feature correlation results lays the groundwork for working with more sophisticated computational models to predict cell viability.

## Computational models can predict cell viability from a combination of kinase inhibition state and gene expression

With our initial analysis of the predictive power of single features from the Klaeger and gene expression data sets completed, we next moved to the development of models that integrated more than one feature with the end goal of predicting cell viability. To do this, we tested four types of models: linear regression, random forest, TabNet and XGBoost. For our initial tests with these models, we used the default settings for all four model types and varied the number of features (either kinase inhibition states or gene expression values) provided to the model. Our cross validation strategy sought to balance our eventual goals of using the resulting models to make predictions about the cell viability effects in new cell lines and in untested compounds. As such, we choose a 10-fold cross validation strategy that randomized fold exclusion across the cell line-compound treatments (63767 total combinations) to improve the likelihood that our model testing results would be similar to downstream experiments. After producing the cross validation splits, we selected a specific number of features and built corresponding models (Fig 3A).

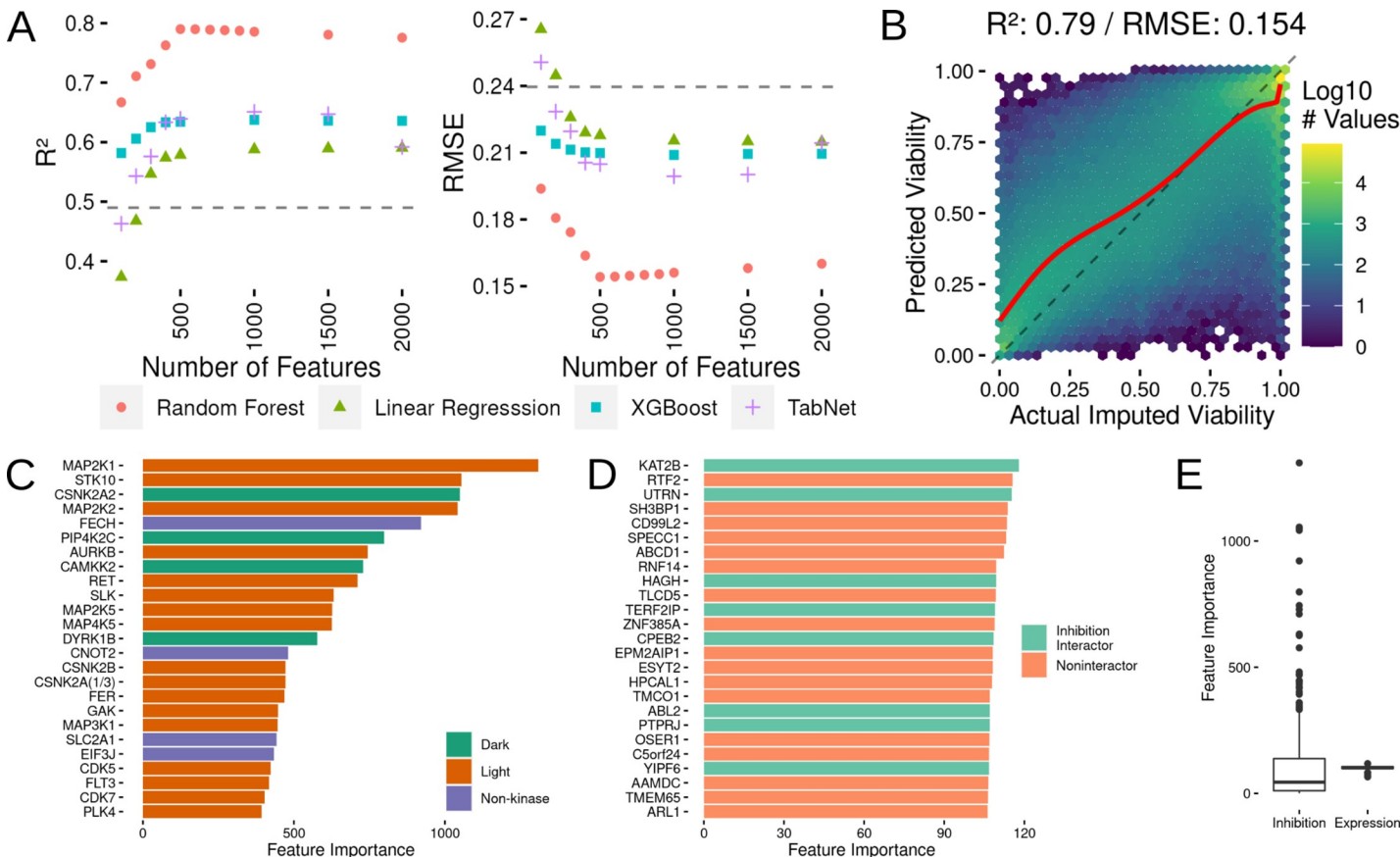

**Fig 3. Development of a Regression Model to Predict Cell Viability and Assessment of Which Features Contribute to Model Predictions.** (A) Comparison of $R^2$ and RMSE values from linear regression, random forest, XGBoost and TabNet models. The gray dotted line shows the performance level of a dose-only model performance. (B) Actual imputed viability versus cross validated model predictions for the random forest model. The dashed line indicates where a perfect set of predictions would appear, while the red line shows a loess fit through the actual results. (C) Variable importance plot for the top 25 features in the final regression model. Each feature is prefixed with act or exp representing either kinase inhibition or gene expression respectively. (D) The top 25 most important expression features in the final model. (E) The overall distributions of feature importance values for the inhibition and expression features.

For benchmarking model performance, we built a naive model that simply used the average cell viability at each of the tested concentrations as a baseline prediction that can be used for comparison (gray dotted lines in Fig 3A), and also compared results to previously run models on similar datasets [31]. Initially, we tested each model type with 100, 200, 300, 400, 500, 1000, 1500 and 2000 features. These preliminary tests showed that the random forest method performed the best at all of these feature counts and that performance ($R^2$ and RMSE) peaked at 500 features and out-performed our baseline dose-concentration-only model. To ensure that we had indeed found the peak in feature performance, we then tested 600, 700, 800 and 900 feature models and found that the 500 feature model was the peak (although all of these models performed very similarly). To better understand this model, we also looked more closely at the predicted versus actual imputed viability of the 500 feature random forest model (Fig 3B). This examination of the cross validation model results, showed that the average model performance was best at higher imputed viability values, while the predictions at lower imputed viabilities were not as accurate. In addition to examining the global model performance we also subsetted the results along compound and cell line results and re-calculated $R^2$ and RMSE (S2A Fig). This result showed that the compound results showed greater variability in $R^2$ as compared to the cell line results, but the RMSE values were similarly distributed.

With random forest using 500 features selected as our best modeling strategy, we moved on to examining feature selection in the cross-validation models as well as parameter tuning. One concern with doing feature selection in each cross validation set was that there would be a large amount of volatility in feature selection between each cross validation model run. We found that in each of the cross validation runs, at least 75% of the features are included in all of the feature selection sets (S2B Fig). To ensure that the default random forest parameter models were near the optimal tuning, we also tested cross-validated models with 1000, 1500 and 2000 trees (500 trees is the default value). Increasing the tree count had little effect on model quality (S2C Fig), so we opted to use the default value of 500 trees. In addition, we also tested the effect of modifying the minimal leaf node size and the number of predictors selected at each branch (S2D Fig) and found minimal effects on $R^2$ and RMSE, so we once again decided to keep the default parameter values.

Our first step in building the final kinase inhibition and gene expression model, was to first select the 500 features that would be included in the model. Using the same correlation ranking scheme used in our cross validated models, 390 out of 520 kinase inhibition states and 110 out 19177 gene expression features were selected for model inclusion. We next built the final random forest model with the full data set and collected variable importance metrics for each of the included features. In order to understand the kinase and non-kinases included in the selected inhibition states, we classified each protein as either a non-kinase or as a well-studied (Light) or understudied (Dark) kinase (Fig 3C) [35]. Several of these genes have well-known roles in cell viability and cancer, including MAP2K1 (MEK1), AURKB and CDK7. Interestingly, the model also identifies several understudied kinases, CSNK2A2, PIP4K2C, CAMKK2 and DYRK1B, as being influential in the model's cell viability predictions. To better contextualize the expression values included in the model, we used the STRING database to see how many of the selected genes interacted with the proteins included in the inhibition features (Fig 4D). Of the 110 genes included in the expression values, 40 interact with at least one protein in the inhibition set and the average expression gene interacts with 1.7 inhibition state genes. In comparison to 10,000 randomly drawn expression gene sets of size 110, 84% interact with fewer than 40 inhibition states and 80% have a lower average inhibition gene interactor count below 1.7. The global view of the variable importance metrics also shows that nearly all of the expression features have similar importance values in the final random forest model (Fig 3E). We next attempted to better understand how the interaction between inhibition states and gene expression levels affected model performance.

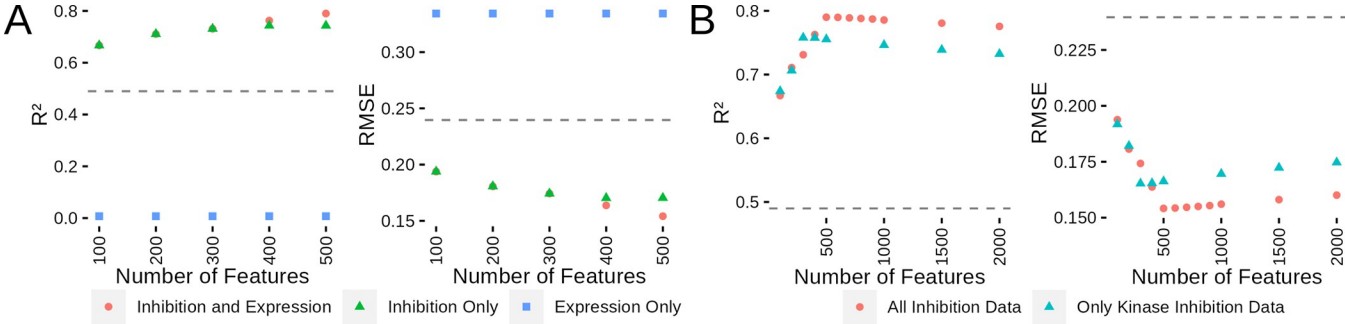

**Fig 4. Model Performance is Best with Access to All Inhibition States and Gene Expression Values.** (A) Comparison of $R^2$ and RMSE performance for models using only expression, only inhibition or inhibition and expression features. (B) Comparison of $R^2$ and RMSE performance for models using gene expression and all inhibition data or only the kinase subset. (C) Plots showing the order of feature selection for the single dose model. (D) Single dose model performance comparison across a range of feature count and with either kinase inhibition state and expression or expression alone.

## The combination of kinase inhibition states and baseline gene expression produces the best predictions

After thoroughly examining the results of the inhibition state and gene expression combined model, we next wanted to investigate how the model would perform when we excluded certain parts of the full data set. Using the same 10-fold feature selection cross validation strategy and the same cross validation fold splits described above, we rebuilt the model using only inhibition state or only gene expression (Fig 4A). The gene-expression-only models performed very poorly ($R^2$ of ~0.01 and RMSE of 0.33), which was expected due to the fact that the gene expression values are fixed and do not vary with the compound concentrations. These model performance differences were also reflected in direct comparisons between individual cell lines and compounds, where none of the expression-only models outperformed the inhibition state only models. When we built models using the inhibition states alone, we observed identical performance for feature counts 300 and below. This was also expected as the correlational feature selection methods always select inhibition features for the first ~350 features. With feature counts of 400 and 500, we observed that the additional information provided by the gene expression features began to improve the model (0.05 improvement in $R^2$ and a 0.02 decrease in RMSE). Thus, while the expression features alone are not sufficient to predict cell viability, they do provide an appreciable improvement in the model performance in combination with inhibition features.

Having established that both inhibition and expression data are needed for the best model performance, we next investigated how the non-kinases in the inhibition data set affected model performance. This question is an interesting avenue to explore as, while the Klaeger et al. study was confined to kinase inhibitors, the presence of ~50% non-kinase proteins inspired us to assess how the model would perform without the non-kinases. We rebuilt the inhibition data set and ran the same modeling methodology including the gene expression values to allow us to compare to our previous models (Fig 4B). The optimum kinase-only inhibition data model had a maximum $R^2$ of 0.76 and a RMSE of 0.17 (compared to $R^2$ of 0.79 and RMSE of 0.15 for the full set). These results indicate that the non-kinases are providing some additional information that the model is able to use, which is in agreement with the presence of non-kinases in the top 25 of the variable importance metrics (Fig 3C).

To further investigate whether the kinase inhibition states are more informative than gene expression values alone, we subset our data to only include the dose for each compound with the highest variation in viability. This is following from a previous publication [36] which built a wide range of models covering chemical and genetic perturbations. By subsetting the data in this fashion, we can more easily compare the relative contributions of kinase inhibition state and gene expression without the variation induced by multiple doses. We used the same feature selection methodology as in the previous section and the shift to only a single dose for each compound generally decreased the kinase inhibition correlations. This allowed more expression values to be included in the model (Fig 4C). After conducting feature selection, we built a set of random forest models with differing numbers of features and found that the kinase inhibition state and expression models outperformed models built with expression data alone. This result demonstrates that even in a more constrained modeling environment, the availability of proteomic based inhibition profiles improves model quality for kinase inhibitors and that the additional information provided by multiple doses can improve modeling results. This is also a limitation though as data comparable to kinase inhibition state does not exist for many classes of compounds, so we view this work as complementary to the broader modeling efforts of Dempster et al. Having fully examined the kinase inhibition state and expression model, we next investigated if any of the other multiomics data sets available could improve upon these models.

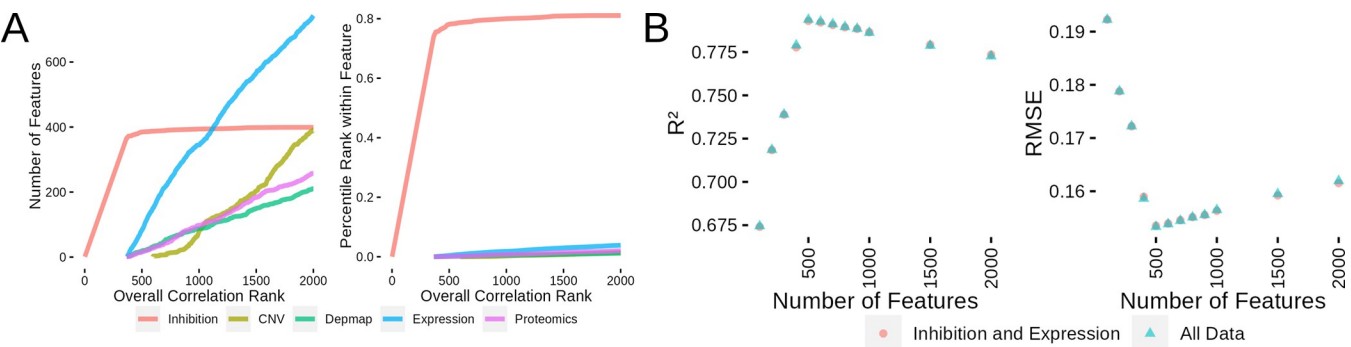

**Fig 5. Regression Models using Additional Data Sets Don't Dramatically Outperform inhibition and Expression Models.** (A) Plot showing the order features are selected for model inclusion (left) and the percentile rank within each feature class as features are selected for inclusion in the model (right). (B) Comparison between models built with inhibition data and expression or all available data sets by $R^2$ (left) and by RMSE (right).

## Models only show mild improvement from inclusion of a broad spectrum of omics data

Gene expression is only one of several different types of comprehensive data that has been collected for many of the cell lines used in the PRISM assay. These additional data sets include:

- DepMap CRISPR-KO screening: genome-wide gene knockout viability measurements (DepMap Score)

- Copy-number-variation: gene level copy number variation (CNV)

- Whole Genome Proteomics: mass spectroscopy-based measurement of relative protein abundance (proteomics)

Given the broad and complementary nature of these data sets, we investigated whether we could integrate these data sets to improve upon the kinase inhibition and gene expression models we described above. The DepMap, CNV and proteomics data sets all overlap with a different number of cell lines present in the PRISM data set (S3A Fig). All of the data sets are available for 212 cell lines (gene expression is available for 476 cell lines represented in PRISM). We focused our modeling efforts on these 212 cell lines to ensure that a complete collection of data was available. We followed the same strategy as in the above modeling effort where we first investigated the correlation between single features and cell viability. The 212 cell line subset showed very similar correlation distributions between kinase inhibition and gene expression (S3B Fig). The newly added feature (CNV, DepMap scores and proteomics) correlations, had correlation distributions very similar to gene expression (S3B Fig). Using the correlation feature ranking, we also determined which features would be included in models of various sizes (Figs 5A and S3C). With these data sets organized and our feature selection techniques specified, we tested how inclusion of these data sets affected model quality.

Based on our previous experience with building the kinase inhibition and expression models, we decided to only test the best-performing random forest method. We also used the same 10-fold cross validation across the cell line/compound combinations. This resulted in higher instability in feature inclusion across the cross validation folds (S3D Fig). As shown in Fig 5B, integration of these other data sets led to performance that was nearly identical to the model with only kinase inhibition and gene expression. The peak performance was achieved at 500 features in both model variants with $R^2$ values of 0.794 (0.153 RMSE) and 0.793 (0.154 RMSE) for the all data and inhibition/expression models respectively. This indicates that gene expression values alone contain substantially similar information as the remaining set of multiomics

data. Given our desire to build a model which uses the most easily reproducible data sets and only minor improvements were observed with the full data collection, we decided to move forward with the integrated model combining kinase inhibition states and gene expression values.

## Validating the models was successful within our ability to replicate previous PRISM results

With the model production decisions finalized, we then applied this model to the untested cell line and compound combinations. The final model was produced using the 63189 cell line and compound combinations with interpolated viability values (Fig 6A). Of the data that went into model production, 476 cell lines and 168 compounds were represented. This left 903 cell lines in the CCLE gene expression data set and 61 Kleager kinase inhibitors that have not been tested in the PRISM viability assays (in addition to a few other untested combinations) where we were able to apply our model to predict cell viability at each of the compound concentrations used in the Klaeger assay. Ultimately, this resulted in us producing predictions for about 250,000 cell line and compound combinations (Sup Data 1). We hope that providing these prediction results will enable other researchers to find interesting or unexpected compounds that target specific cancer types. For the work presented here, we focused our validation efforts on a subset of breast cancer cell lines.

Our first goal when beginning to validate a subset of model predictions was to see how well we could replicate the results from the PRISM assay. We selected the well characterized triple negative breast cancer (TNBC) cell line HCC1806 and a set of compounds that displayed a range of viability effects from the 134 Kleager kinase inhibitors that had been used in the PRISM assay with the HCC1806 cell line (Fig 6B). Several of these compounds performed very similarly in our assay as compared with the imputed viability PRISM values, notably Cobimetinib, UCN-01, AT-9283, Lestauritinib and Dinaciclib. However, several of the compounds that showed high viability effects at high concentrations were not reflected in the imputed viability results, which lowered the replication $R^2$ to 0.492 and the RMSE to 0.299 (Fig 6C). To put these replication efforts in context, we looked for experimental cell viability results from the NCI-60 [37] and PRISM results where the same compound and cell line were assayed. In order to gain a broader understanding of cell line viability replicability, we included every compound match we could find between the two data sets. We found 172 compounds and 32 cell line matches between these data sets and found an overall $R^2$ of 0.444 and an RMSE of 0.296 (S4 Fig). These results were in agreement with our much smaller PRISM replication effort, indicating that the variance between model predictions and experiments is no worse than the variance observed between experiments replicated by different groups. With the inherent limitations identified by the replication effort acknowledged, we next moved into testing new cell line and compound combinations.

We started testing new cell line and compound combinations by continuing with the HCC1806 line and adding in the HER2 positive breast cancer cell line BT-474. We selected a set of compounds predicted to have a range of effects on the two cell lines and then conducted a viability screen with each of these compounds (Fig 6D). Much like the replication attempt, we observed several compounds where the predicted viabilities were close to the measured viability (K-252a, UCN-01, PF-3758309 and Lesauritinib). Overall, the $R^2$ (0.518) and RMSE (0.239) values were comparable with replication effort, indicating that the model was performing well on new compounds (Fig 6E). As our most challenging final test, we decided to test two cell lines that are not present in the PRISM data set (HER2+ line SKBR3 and TNBC line SUM159PT) against a set of compounds that weren't included in the PRISM compound set. Once again, with this "double-untested" experiment, we selected a set of compounds predicted

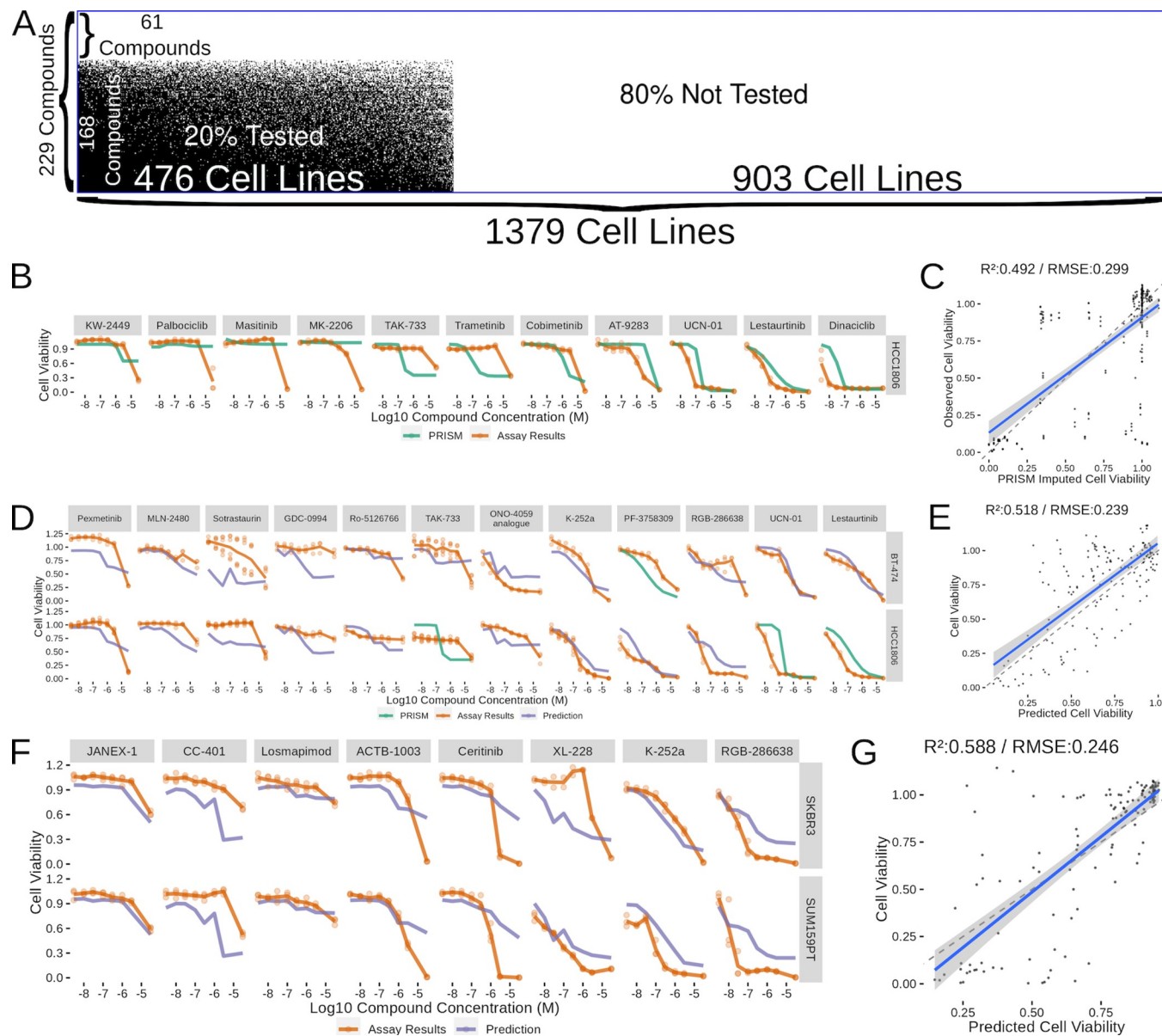

**Fig 6. Validating a Subset of Compound Predictions in Breast Cancer Cell Lines.** (A) Visualization of the space of compound (Y-axis) and cell line (X-axis) combinations that have been tested (white) and non tested (black) with a blue box surrounding the entire visualization. (B) Cell viability results from testing a set of compounds (labeled above each curve) and a cell line (HCC1806) already tested in the PRISM collection. (C) Scatterplot summarizing all the results from part B into a single plot with a linear best fit line showing in blue. (D) Cell viability results and corresponding predictions or PRISM results from a set of cell lines included in PRISM (BT-474 and HCC1806) and a selection of compounds which were mostly not included in PRISM. (E) Scatterplot summarizing all the prediction results from part D into a single plot. (F) Cell viability results and corresponding predictions for a set of cell lines and compounds not included in PRISM. (G) Scatterplot summarizing all the results from part F into a single plot.

to have varying effects across concentrations and observed a combination of compounds with strong and weak correlation between predictions and results (Fig 6F). Notable among the better results were JANEX-1, Losmapimod and K-252a, while the model struggled with CC-401 and parts of the RGB-286638, ACTB-1003 and Ceritinib curves. The overall performance of the model ($R^2$ of 0.588 and RMSE of 0.246) were comparable to the other model validation results (Fig 6G). These independent validation efforts demonstrate that the model predictions are able to generalize into previously untested cell lines and compounds.

## Discussion

Given the potential of targeted kinase inhibitor therapies, the ability to predict how a given treatment may alter kinome state and lead to a given phenotype is fundamentally enabling. In this work, we developed a set of computational models that predict cell viability after treatment with a set of small molecule kinase inhibitors. To accomplish this, we used several publicly available data sets that provided information concerning the untreated gene expression of the cell lines used in the viability screen and another that gave detailed information about the proteins targeted by small molecule kinase inhibitors. We examined how single gene expression and kinome state values were related to cell viability and how models with various numbers of gene expression and kinome state values varied in quality. In addition to gene expression, we also tested a set of models which included a broader range of baseline measurements (CNV, proteomics and gene essentiality) and concluded that these additional data sets were not able to significantly improve model performance. Finally, we tested some of the model predictions in several triple negative and HER2 positive breast cancer lines and found acceptable agreement between the model predictions and experimental results.

This work demonstrates how knowledge of the inhibition state of the kinome, derived from a proteomic assay based on a four cell lysate mixture, can predict a cellular process as fundamental as viability. Importantly, the models achieved these surprising results by using a "generic" or "general" kinase inhibition profile measured with proteomic kinobead profiling of a four cell line lysate exposed to an extensive library of kinase inhibitors at multiple doses [26]. Thus, the models learned by linking non-cell line specific kinome inhibition state information with that of specific drug-cell line relationships.

We acknowledge several limitations of this work. First, all of the results in this paper rest on the availability of kinome profiling data specific to a given kinase inhibitor, so the methods here are not applicable to prediction of cell viability effects in any other class of compound. We believe that a similar strategy could be used to build models in compound classes where the spectrum of targets were as comprehensively identified. The universe of small molecule kinase inhibitors is substantially larger than those that were surveyed by Klaeger et al., but since our modeling methodology depends on the comprehensive nature of their work, we're limited in the number of compounds where we can make predictions. One of our next goals is to attempt to broaden the scope of compounds through integration of other high-content kinome profiling techniques such as KinomeScan and Nano-BRET. In addition, while the models described in this paper do make somewhat accurate predictions, these results point to a degree of missing predictability in cell viability for which new methods and data will need to be developed and collected. Also, since this work has targeted building a single comprehensive model, it is likely that subtle cancer type specific relationships are not captured such as the relationship between RXRG expression and melanoma [36]. This can be addressed by subsetting the model to make predictions about specific cancer types/subtypes. There is also an extensive set of alternative hyperparameter settings and potential modeling methodologies that we did not explore in this work. We also hope that by providing a full set of viability predictions for the broad range of cancer cell lines covered by the CCLE that this work can act as a resource for other researchers to find unexpected or interesting kinase inhibitors that affect their most used cell line model systems.

This work also suggests several extensions that would broaden or improve the model. Given recent interest in finding new compound combinations computationally, we are beginning to examine how best to combine the information from multiple compound kinome inhibition states to predict the resulting cell viability effects. This would allow us to run computational drug combination screens. In addition, the methods outlined here will also

likely work for any phenotype that can be measured after treatment with small molecule inhibitors and with sufficient throughput to gather a large enough data set. Finally, while we have made all of the code and data necessary to reuse our models available to the public on github, we also acknowledge that this is not the most user-friendly method for allowing non-computationally minded users to access the model. Thus, we also plan on developing a web-based system for allowing non-computationally minded users to submit a gene expression profile and receive a set of predictions concerning how their cellular system is expected to respond to the Klaeger set of kinase inhibitors.

Overall, we hope that this paper makes a contribution to our understanding of how the overall state of kinome in response to small molecule inhibitors contributes to cell viability phenotypes. Our findings demonstrate that while individual kinase inhibition states and other single gene or protein readings are not very predictive of cell viability, machine learning approaches are able to combine sets of measurements related to the small molecule kinase inhibitors and gene expression data to make cell viability predictions. The results presented here show how a thorough understanding of kinase activity levels in conjunction with baseline omics data can be used to gain a better understanding of phenotypes such as cell viability.

## Methods

Our methods can be divided into two parts describing the computational aspects of this work and the experimental methods used to test the output of the computational components.

### Data sources

We used two primary data sources for this paper: the supplemental data section from Klaeger et al.[26] and the cell viability screening results from the PRISM lab. Specifically, we collected and organized the kinase inhibition states from supplemental Table 2 of Klaeger et al, focusing on the Kinobeads subsheet. As for the PRISM data, we used the data from 2019 Q4 (labeled 19Q4 in the depmap portal), specifically the secondary screening data. In addition to these two data sets, we used supplemental data sets from the CCLE [34] and DepMap [38]. These data included results from baseline RNAseq (CCLE_expression.csv), copy number variation (CNV, CCLE_gene_cn.csv) and CRISPR-KO viability screening (CRISPR_gene_effect.csv). The 2021Q3 versions of these files were used. The proteomics data was downloaded from the Gygi lab website (https://gygi.hms.harvard.edu/publications/ccle.html), specifically Table S2 [39]. We also used version 11.5 of the STRING [40] protein network database (9606.protein.links.v11.5.txt.gz).

### Data preprocessing

The scripts implementing these descriptions are all available through github.

### Klaeger et al. kinase inhibition profiles

We read the values from the supplemental data table into R and produced a list of all proteins observed in any of the kinase inhibitor treatments. Since this table only contains the proteins affected by each compound, we filled in the relative intensity values for genes not associated with a given inhibitor with the default value of 1. There was a small (1.8%) number of single concentration values missing from the listed affected proteins, so we filled these values as the average of two nearest concentrations. Finally, a smaller set (0.01%) of likely outlier relative intensity readings were truncated to the 99.99 percentile (3.43).

## PRISM cell viability

Since relatively few of the concentrations used in the PRISM assay match those used by Klaeger et al., we opted to use the response curve parameters provided through the depmap portal to interpolate the cell viability values. We interpolated these values at 30 μM, 3 μM, 1 μM, 300 nM, 100 nM, 30 nM, 10 nM and 3 nM to match those used by Klaeger et al. We applied a filter to remove any response curve parameter set that indicated that a given cell line and compound combination produced enhanced cell growth with increasing compound concentration. To perform the viability extrapolation, we used the four-parameter log-logistic formula described in the drc R package [41].

## Gene expression, CNV, CRISPR-KO and proteomics

The files provided by the depmap portal for gene expression, CNV and CRISPR-KO values required very little modification to work in our machine learning pipelines. The primary modification was to add identifiers to each gene label, to ensure that omics data related to the same gene weren't accidentally combined. The CRISPR-KO data also required an additional filter to remove 10 cell lines with missing data. The proteomics data processing was slightly more complicated, as there were substantially more protein readings missing from many more lines. In the cases of missing protein readings, we imputed these values to the minimum value for the overall distribution of that protein minus one standard deviation.

## String

The STRING database [40] also required only mild preprocessing to extract the proteins that interacted with the components of our models. We filtered the interaction list to the high confidence (above 0.7) set and used bioMart [42] to convert the Ensembl protein identifiers to HGNC identifiers for matching with the other data sets.

## Modeling techniques and types

To assess our models we used a 10-fold cross validation strategy which randomized training and test set inclusion across the cell line and compound combinations. Thus, for any given viability curve resulting from treatment of a cell line with a compound, all of the results from the assay were considered as one unit for cross validation purposes. All steps of feature selection were also conducted under this cross validation framework as well. For every fold of our data, we recalculated the correlation coefficient between cell viability and the features available to the model (kinase inhibition state, gene expression, etc) using only the data in the training set. The number of features was varied as specified in the results section. We used the entire data set to build the final model used to make the predictions in S1 Table and the results displayed in Fig 6.

   We used random forest, XGBoost, TabNet and linear regression for all of our modeling efforts. All of our models are implemented using the tidymodels framework in R. We used the ranger random forest engine [43], the default XGBoost engine [44] and the default ordinary least squares linear regression engine. For all of our initial testing of these models we used the default single set of hyperparameter settings to narrow our search for an acceptable model. This search indicated that the random forest model performed the best, so we attempted to further tune three additional parameters, the number of trees, the number of selected predictors and the minimal node size across the following ranges:

- Number of Trees: 500 (default), 1000, 1500 and 2000

- Number of Predictors: 11, 22 (default for 500 tree model), 33 and 44

- Minimal Node Size: 3, 5 (default) and 10

## Compound testing

BT-474, HCC1806, SUM-159 and SKBR-3 cells were grown in ATCC recommended media and seeded at 4000, 2000, 4000 and 500 cells per well respectively, in white flat-bottom 96-well plates (Corning). 24 hours after seeding, cells were treated with the respective drugs prepared in DMSO. All drugs were dosed at the same eight concentrations used in the Klaeger study: 30 μM, 3 μM, 1 μM, 300 nM, 100 nM, 30 nM, 10 nM and 3 nM. Seventy-two hours post-treatment, cells were lysed with CellTiter-Glo (Promega) per the manufacturer's protocol. Luminescence was read using the PHERAstar FS microplate reader (BMG Labtech) and gain adjustments were conducted for each cell line. Data were normalized row-wise to the DMSO-only (0.5% on cells) control samples on each plate to calculate relative viability. Quality checks were performed to look at the data distribution and the presence of spatial bias on a plate. A quality control metric of <120% of DMSO was applied to all rows analyzed. Across all >150 rows analyzed, only one row of XL-228 treated SKBR-3 cells failed to meet this criteria and was removed from analysis.

## Supporting information

**S1 Fig. Expanded Correlation Rankings (Associated with Fig 2).** Extended version of Fig 2D covering all correlation ranks.
(TIFF)

**S2 Fig. Feature Selection with Cross Validation and Assessment of Increasing Random Forest Trees (Associated with Fig 3).** (A) The effect of cross validation data division on which features are selected for model inclusion. (B) The effect on $R^2$ and RMSE of increasing the number of trees used in the random forest algorithm. (C) The effect on $R^2$ and RMSE of modifying the selected predictor count and the minimal node size used in the random forest algorithm. (D) The distribution of $R^2$ and RMSE for single compound or cell line cross validation results.
(TIFF)

**S3 Fig. Expanded Correlation Rankings and Effect of Cross Validation Subsetting on Feature Selection (Associated with Fig 5).** (A) Upset plot showing the overlap between data sets across cell lines in the PRISM assay. (B) Small multiples plot showing the correlation of individual features to imputed cell viability for each of the feature types considered in this model. (C) Full feature correlation rankings for all data set types considered for Fig 5. (D) Effect of random 10-fold cross validation subsetting on which features are included in what percentage of the cross validation data sets.
(TIFF)

**S4 Fig. Comparison of Cell Line and Compound Matched PRISM and NCI-60 Viability Results.** The red line shows a Loess fit through the data set.
(TIFF)

**S1 Table. Full Viability Curve Predictions for CCLE Cell Lines.**
(XLSX)

## Acknowledgments

We would like to thank Donglin Zeng for helpful discussions concerning feature selection techniques and UNC Research Computing for access to the computational resources necessary for this work. We would also like to thank Madison Jenner and Jen Jen Yeh for providing a sample of JANEX-1.

## Author Contributions

**Conceptualization:** Matthew E. Berginski, Chinmaya U. Joisa, Shawn M. Gomez.

**Formal analysis:** Matthew E. Berginski, Chinmaya U. Joisa, Shawn M. Gomez.

**Software:** Matthew E. Berginski.

**Validation:** Chinmaya U. Joisa, Brian T. Golitz.

**Writing – original draft:** Matthew E. Berginski, Chinmaya U. Joisa, Shawn M. Gomez.

**Writing – review & editing:** Matthew E. Berginski, Chinmaya U. Joisa, Shawn M. Gomez.

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
