## [Decision Letter · Decision Letter 0]

3 Aug 2022

Dear Professor Gomez,

Thank you very much for submitting your manuscript "Kinome Inhibition States and Multiomics Data Enable Prediction of Cell Viability in Diverse Cancer Types" for consideration at PLOS Computational Biology.

As with all papers reviewed by the journal, your manuscript was reviewed by members of the editorial board and by several independent reviewers. In light of the reviews (below this email), we would like to invite the resubmission of a significantly-revised version that takes into account the reviewers' comments.

We cannot make any decision about publication until we have seen the revised manuscript and your response to the reviewers' comments. Your revised manuscript is also likely to be sent to reviewers for further evaluation.

Sincerely,

Inna Lavrik

Associate Editor

PLOS Computational Biology

Feilim Mac Gabhann

Editor-in-Chief

PLOS Computational Biology

Reviewer's Responses to Questions

**Comments to the Authors:**

Reviewer #1: The recent development of cancer treatment emphasizes targeted therapies. Protein kinases represent one of the largest classes of anti-cancer targeted therapies. Therefore, uncovering the correlation between kinome activity and cancer cell viability via regression model could provide a list of potent drugs based on predicted cell viability value ahead of treatment and infer potential key kinases in regulating cell viability. Berginsiki et al set out to determine whether a combination of kinase inhibition profiles at multiple doses and gene expression could predict the effect of kinase inhibitors on cancer cell viability. They integrated kinase inhibition profile data and gene expression data of different cancer cell lines to build a random forest model to predict cell viability. The final model achieved R2 value as 0.79 in 10-fold cross-validation. This manuscript compared linear and non-linear models, utilized feature ranking via Pearson’s correlation, and introduced data from various aspects like copy number variation and proteomics as additional input for model training. Besides developing the model, this manuscript showed some insights from the model itself, pointing out that most top-ranking features are from kinase inhibition data and that genes selected as model features have a high possibility of interacting with kinase. This manuscipt also considered the experimental variation and evaluated the difference between the original PRISM value and lab assay result before validating the model results on new compound–cell line combinations.

Major comments

1. The conceptual bias of training data. The kinase inhibition profiles comes from Klaeger et al, whose data is derived from a mixture of four cell lines, while cell viability and gene expression data (PRISM) are from a panel of over 400 cancer cell lines. The combination of kinase inhibition state from a mixture of cell lines and cell line-specific gene expression data might not be a proper input to train a model predicting individual cell line viability. Have the authors tried to use kinase inhibition profiles from gold standard radioactive assays which could avoid in vivo bias due to cell line variation? For example, GSK kinase inhibitor profiling or HMS LINCS profiling done at multiple doses.

2. Lack of innovation from a modeling perspective. To build the model, Berginsiki et al applied linear regression, random forest, and XGBoost with a limited parameter tuning process. Introducing a neural network with different transformers might improve the performance for large-scale multi-dimensional data. To optimize the random forest model, there should be parameters other than tree number to adjust, like subsample size, max leave, depth and child weight, etc. Show optimization of RF. Is it possible to make some more improvements to this model if including these features in parameter tuning, such as using grid search or randomized search for 1000 iterations or so?

3. I applaud the author's effort to examine the difference between original PRISM data and lab assay results, and there exists a similar trend between model prediction curves and lab assay results, however, R2 score of 0.5 is merely reasonable. Moreover, this comparison is based on the assay results and the model prediction values, which could not infer how the model might perform on the original PRISM data, given the observed difference between PRISM value and assay result.

Reviewer #2: Summary:

In this study, the authors develop a set of predictive models for cell viability across varying concentrations of kinome inhibitors, using kinome "inhibition states" as predictive features. The work builds on a 2017 paper by Klaeger et al. which profiled 243 kinase inhibitors to identify potential off-target/polypharmacology effects. This study integrates the kinome profiles from Klaeger et al. with -omics data from CCLE and PRISM, showing that kinome profiles tend to be more informative in viability prediction than baseline gene expression values, and additionally that kinome profiles and gene expression can be combined to produce slightly better models than either data type alone. The authors validate their modeling strategy on held-out kinase inhibitors and cell lines, showing that for a subset of breast cancer cell lines their models tend to generalize well.

The paper is clear and well-written, and the experiments provide compelling evidence that kinome inhibition profiles contain predictive signal for cell viability screening of kinase inhibitors. The validation on breast cancer cell lines and the predictions across all of PRISM will serve as a useful resource, and the GitHub repository containing the code is well-documented and easy to navigate. However, there are some aspects that could be explored further to ensure the robustness of the study's main conclusions, particularly with relevance to existing work in viability prediction.

Major comments:

1. The conclusions of the authors in this study seem slightly at odds with previous work, particularly Dempster et al. 2020 (https://doi.org/10.1101/2020.02.21.959627) which found that baseline expression was generally effective for predicting cell viability for various types of perturbations, moreso than genomic profiles. Importantly, the studies are looking at different datasets and drug classes: the Dempster et al. study looked at a broader set of compounds rather than just kinase inhibitors, and at GDSC and several other datasets in addition to PRISM. However, the application to cell viability and overall conclusions of both studies still seem quite related.

Given the seemingly contradictory conclusions of these two studies, I think the question I would be most interested in is whether this is a biological difference (i.e. kinase inhibitors behave differently than the larger set of compounds as a whole), or whether a difference in experimental design is causing the discrepancy (e.g. model setup, label definition, cross-validation splitting strategy, etc, all of which have subtle differences between the two studies). Glancing through the methods of both papers I noticed a few differences - perhaps most notably the Dempster et al. study uses only the PRISM dose with the most variance over cell lines for each compound, while the present study looks at multiple doses as described in the Methods section and in Klaeger et al.

It would be informative if the authors tried running their gene expression pipeline on their set of kinase inhibitors with the labeling strategy (selecting a single representative dose) described in Dempster et al. - if the results are unchanged, it would inspire more confidence that the set of kinase inhibitors studied in this paper are indeed biologically different from the rest of the compounds tested in the Dempster et al. study, rather than reflecting an experimental design difference. If the results are different, this would also be a noteworthy outcome - it would support the idea that looking at multiple doses, rather than picking a single representative dose, is a necessary/important difference between the studies.

2. Related to the first point, the example of BRAF inhibitors in Figure 6 of the Dempster et al. paper (dabrafenib and vemurafenib) could be a useful positive control, or example to use to examine the differences between the studies. Their data suggest that baseline RXRG expression, particularly in melanoma cell lines, is a strong univariate predictor of response to dabrafenib and vemurafenib - is that still the case for the authors' pipeline (i.e. are BRAF inhibitors a class of drugs where gene expression is particularly effective, and the poor performance shown in Figure 4 is mostly driven by other cell lines or kinase inhibitors), or are the results different when the labels for multiple doses are introduced? Does the authors' multivariate gene expression-based model perform well for BRAF inhibitors?

3. Many of the figures in the paper (e.g. Figure 4, Figure 5) show results/performance metrics summarized across all cell lines and compounds in the dataset. This makes sense for a summary figure, but it seems to me that this could potentially be obscuring variation between cell lines or between compounds. For instance, despite the fact that gene expression does not perform well for viability prediction in general (shown in Figure 4), are there any compounds or any cell lines/tissues of origin where gene expression does outperform kinase inhibition data? If so, is there any biological interpretation for the variation? If this is not the case (i.e. gene expression performs uniformly poorly across cell lines and tissues of origin, and/or kinase inhibition profiles perform uniformly well) this would also be interesting to note.

4. Overall the paper is very clear and well-written. However, as a reader, I would appreciate a bit more clarification earlier on in the paper about exactly what a "kinome inhibition profile" is in the introduction/summary figure since this is central to understanding the paper, especially given that this is a general computational biology journal with readers from diverse fields. Reading through the Klaeger et al. paper, this quote seems like a good summary of why kinome inhibition profiles are useful:

"Owing to the fact many compounds target the structurally and functionally conserved adenosine 5′-triphosphate (ATP)–binding site, polypharmacology (that is, drugs that act on more than one target) is commonly observed. Target promiscuity may have advantageous or detrimental therapeutic consequences."

The idea seems to be that kinase inhibitors can be either targeted or more general, but almost all of them have some "off-target" or polypharmacology effects. Knowing the empirical profile of kinase inhibition for a drug, then, can give you more information than just knowing what it's designed to target. It would be good to put some form of this summary in one of the introductory paragraphs, and/or to explicitly mention exactly what the model features represent (my understanding is that the input is a drug/perturbation x kinase matrix, where each entry quantifies binding/drug interaction strength for the given drug and kinase) since many readers might not be immediately familiar with the data format, or with the observation that kinase inhibitors do not necessarily bind specifically to their intended targets.

Minor comments:

5. The authors should be more specific about what form of linear regression is being used in the Methods section. Does the tidymodels default use any form of regularization (e.g. a ridge/LASSO/elastic net penalty), or is it an ordinary least squares fit? It may also be good to describe the default hyperparameter selection strategy for the random forest and XGBoost models - does tidymodels test a variety of parameters using some kind of internal cross-validation, or does it choose a single set of default parameters? Is there any particular reason the authors chose to tune the number of trees in the random forest model and not any other hyperparameters? Hyperparameter selection details can have a considerable effect on performance, at least in my experience, and it is useful to be clear about what was explored/settled on for the final models and why.

6. Methods section, page 18 under "STRING": Ensemble -> Ensembl

**Have the authors made all data and (if applicable) computational code underlying the findings in their manuscript fully available?**

Reviewer #1: Yes

Reviewer #2: Yes

PLOS authors have the option to publish the peer review history of their article (what does this mean?). If published, this will include your full peer review and any attached files.

Reviewer #1: No

Reviewer #2: No
---

## [Decision Letter · Decision Letter 1]

20 Jan 2023

Dear Professor Gomez,

We are pleased to inform you that your manuscript 'Kinome Inhibition States and Multiomics Data Enable Prediction of Cell Viability in Diverse Cancer Types' has been provisionally accepted for publication in PLOS Computational Biology.

Best regards,

Inna Lavrik

Academic Editor

PLOS Computational Biology

Feilim Mac Gabhann

Editor-in-Chief

PLOS Computational Biology

Reviewer's Responses to Questions

**Comments to the Authors:**

Reviewer #2: The comments from my previous review have been addressed - thank you for the great work. In particular, the experiments showing that kinase inhibitor profiles still provide the most predictive signal even when only a single representative (highest variability) dose is used emphasize the robustness of the results, and the added analyses exploring per-compound and per-cell line performance variation is another interesting aspect of the revised paper.

I have a few small comments/suggestions on the revised version of the manuscript:

1. One aspect of the response to reviewers that stood out to me was this: "We think of the kinase inhibitor profiles as providing a cell-line agnostic initial “state” that is expected to be induced by inhibitor treatment, with the RNA expression altering that state in a cell-line-specific manner (and thus the improvement in predictions with its inclusion)." I thought this was an interesting way to think about the findings in the paper - i.e. the kinase inhibition data provides information about each compound or perturbation, and the gene expression/omics data provides information about the cell line. Since the kinase inhibition profiles were so valuable in terms of predictive signal, I wonder if there are there any other data sources that could provide information about each compound in a cell-line-general manner similarly to, or complementarily to, the kinase inhibition profiles: maybe information on chemical structure, where applicable (e.g. Morgan fingerprints or some other kind of molecular embedding) could provide a performance improvement as well?

If so (or if there are other examples I'm not aware of) this could be a useful addition to the discussion section. To be clear, the findings in the paper are valuable in their own right and adding more compound-specific information is probably beyond the scope of this study, but mentioning it as a future direction or an existing area of research could help to put the study in a broader context.

2. In the first sentence of the revised abstract, should "compounds that inhibit kinase activity emerging as a primary focus" be "compounds that inhibit kinase activity *are* emerging as a primary focus"?

**Have the authors made all data and (if applicable) computational code underlying the findings in their manuscript fully available?**

Reviewer #2: Yes

PLOS authors have the option to publish the peer review history of their article (what does this mean?). If published, this will include your full peer review and any attached files.

Reviewer #2: No

---

## [Editor Report · Acceptance letter]

15 Feb 2023

PCOMPBIOL-D-22-00783R1 

Kinome Inhibition States and Multiomics Data Enable Prediction of Cell Viability in Diverse Cancer Types

Dear Dr Gomez,

I am pleased to inform you that your manuscript has been formally accepted for publication in PLOS Computational Biology. Your manuscript is now with our production department and you will be notified of the publication date in due course.

With kind regards,

Timea Kemeri-Szekernyes
